# Effect of robotic-assisted gait training on gait and motor function in spinal cord injury: a protocol of a systematic review with meta-analysis

Lei Wang [1], Jin-lin Peng,[2] Ai-lian Chen[1]

¹Department of Rehabilitation Medicine, Hunan Provincial People's Hospital, Changsha, Hunan, China
²Tongji Hospital of Tongji Medical College of Huazhong University of Science and Technology, Wuhan, Hubei, China

**Correspondence to**
Professor Ai-lian Chen;
652326303@qq.com

## ABSTRACT

**Introduction** Robotic-assisted gait training (RAGT) has been reported to be effective in rehabilitating patients with spinal cord injury (SCI). However, studies on RAGT showed different results due to a varied number of samples. Thus, summarising studies based on robotic-related factors is critical for the accurate estimation of the effects of RAGT on SCI. This work aims to search for strong evidence showing that using RAGT is effective in treating SCI and analyse the deficiencies of current studies.

**Methods and analysis** The following publication databases were electronically searched in December 2022 without restrictions on publication year: MEDLINE, Cochrane Library, Web of Science, Embase, PubMed, the Cochrane Central Register of Controlled Trials and China National Knowledge Infrastructure. Various combinations of keywords, including 'motor disorders', 'robotics', 'robotic-assisted gait training', 'Spinal Cord Injuries', 'SCI' and 'gait analysis' were used as search terms. All articles on randomised controlled trials (excluding retrospective trials) using RAGT to treat SCI that were published in English and Chinese and met the inclusion criteria were included. Outcomes included motor function, and gait parameters included those assessed by using the instrumented gait assessment, the Berg Balance Scale, the 10-m walk speed test, the 6-min walk endurance test, the functional ambulation category scale, the Walking index of SCI and the American Spinal Injury Association assessment scale. Research selection, data extraction and quality assessment were conducted independently by two reviewers to ensure that all relevant studies were free from personal bias. In addition, the Cochrane risk-of-bias assessment tool was used to assess the risk of bias. Review Manager V.5.3 software was used to produce deviation risk maps and perform paired meta-analyses.

**Ethics and dissemination** Ethics approval is not required for systematic reviews and network meta-analyses. The results will be submitted to a peer-reviewed journal or presented at a conference.

**PROSPERO registration number** CRD42022319555.

## INTRODUCTION

Spinal cord injury (SCI) is a serious disabling disease that often causes paraplegia or quadriplegia and affects patient's sensory, motor and autonomic nervous functions.[1][2] SCI leads to various complications such as pressure ulcers, lung infections and urinary tract infections.[3] It also affects patients' quality of life and living standard and imposes a heavy burden on families and society.[4] It ultimately shortens patients' life expectancy.[5] In addition, the mortality rate of patients with SCI is higher than that of the general population.[6–8] National statistical data show an increasing incidence rate of SCI annually, and that the incidence rate of SCI per million residents is 9.3 persons/year.[9] During the rehabilitation treatment of SCI, improving the walking ability, self-care ability and self-esteem of patients is an important aspect that helps them return to society and reduces their costs. Therefore, increased exercise capacity of the lower limbs is crucial to daily independence and social reintegration for this population, which mainly functions in standing and walking.[10][11]

Robot-assisted gait training (RAGT) can improve the walking ability,[12] lower limb strength and independence of patients with incomplete SCI.[13] RAGT can also improve balance function and has been gradually

---

### STRENGTHS AND LIMITATIONS OF THIS STUDY

⇒ This study was the first meta-analysis to systematically evaluate the efficacy and safety of robotic-assisted gait training in the treatment of spinal cord injury (SCI).

⇒ The results of this study provided evidence for the treatment of patients with SCI and helped therapists and patients to choose appropriate treatment methods.

⇒ Two reviewers independently conducted research selection, data extraction and quality assessment to ensure that all relevant studies were free from personal bias.

⇒ The language categories of the research search were only included in English and Chinese, and the final search results would have some bias.

applied in patients with SCI.[14] In patients with SCI, robots for lower limb rehabilitation can effectively and safely improve walking ability; reduce pressure ulcers,[15] lung infections,[8] urinary tract infections and other complications[16]; improve dignity; and reduce costs. However, high-quality evidence-based medical studies that systematically evaluated the efficacy of RAGT in the treatment of SCI remain scarce.

Therefore, summarising studies based on RAGT-related factors is critical for the accurate estimation of the effects of RAGT on SCI. This meta-analysis aims to systematically evaluate the efficacy of RAGT in alleviating motor dysfunction and restoring speech ability in patients with SCI based on randomised clinical trials (RCTs), find strong evidence demonstrating that using RAGT is effective in the treatment of SCI and analyse the deficiencies of current studies.

## METHODS

The protocol of this systematic review was planned and conducted following the Preferred Reporting Items for Systematic Reviews and Meta-Analyses (PRISMA) Protocols guideline and PRISMA 2020 guidelines and was performed following a protocol registered in PROSPERO.[17 18] The plan starts on 1 March 2023 and ends on 1 June 2023. The review process is shown in figure 1.

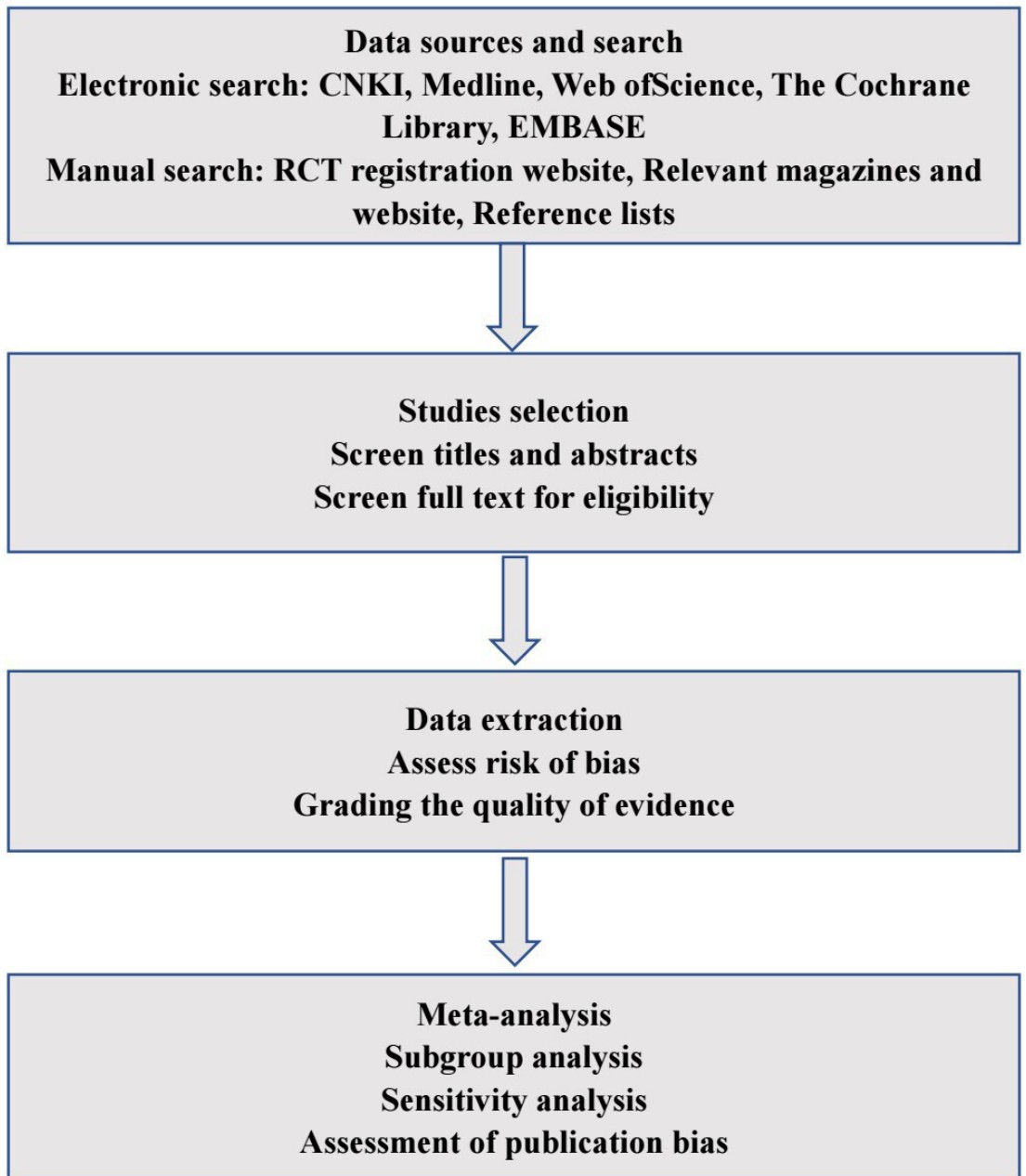

**Figure 1** Flow chart of meta-analysis for robotic-assisted gait training in patients with spinal cord injury. CNKI, China National Knowledge Infrastructure; RCT, randomised clinical trial.

## Search strategy

Two reviewers (J-LP and LW) electronically searched the following publication databases in December 2022 without restrictions on publication year: MEDLINE, Cochrane Library, Web of Science, Embase, PubMed and China National Knowledge Infrastructure. Various combinations of keywords, including 'motor disorders', 'robotics', 'robotic-assisted gait training', 'Spinal Cord Injuries', 'SCI' and 'gait analysis' were used as search terms. The key terms matched the appropriate Medical Subject Heading terms. Pre-searches were performed. Then, the final search was conducted, and relevant journals and references of review articles were manually searched online to identify papers that may have been missed in the electronic database searches.

## Eligibility criteria

### Inclusion criteria

(1) Study design: Only RCTs were included. (2) Selected population: Participants diagnosed with SCI, namely, individuals with any level of traumatic SCI, regardless of the time since injury, sex and age were included. (3) Type of intervention: The experimental groups received RAGT or RAGT combined with other physical therapies. The control group did not receive RAGT or received other types of physical therapy. (4) Comparison: The treated subjects were compared at baseline and then with the control or sham-stimulated subjects. (5) Type of outcomes measured: Gait analysis indicators, including gait speed (m/s), step length (cm), double support phase (% walking cycle), single support phase (% walking cycle) and symmetry index; Berg Balance Scale; American Spinal Injury Association assessment scale; Holden walking ability classification (functional ambulation category scale); 10 m walk speed test; 6 min walk endurance test; and Walking Index for Spinal Cord Injury II score.

### Exclusion criteria

Studies involving animal research, conference research, protocol studies or computer model research and duplicate papers were excluded. Two reviewers (J-LP and LW) independently screened titles and abstracts to identify articles reporting studies that met the inclusion criteria. Then, the full-text versions of the identified articles were obtained and separately screened to ensure that they met the inclusion criteria. Moreover, a third reviewer (ALC) made the final assessment regarding whether or not full-text papers met the inclusion criteria.

## Data extraction

A reviewer (LW) prepared the general information and data collection process by another reviewer (J-LP). The format of data collection included research design, participants (number, diagnosis, age and target population numbers in each group), eligibility criteria, intervention used on the research group and control group (ie, site of stimulation, intensity, number of sessions and time of each session) and outcomes of interest.

## Quality assessment

The quality evaluation of the included studies was performed independently by two reviewers (J-LP and LW) and was revised by the third reviewer (A-LC). The methodological quality of the intervention studies was assessed using the Physiotherapy Evidence Database (PEDro) scale. The PEDro scale is a valid and reliable measure of the methodological quality of RCTs. This 10-item scale is based on the core criteria for RCT quality assessment.[19] The quality of papers was classified based on the PEDro scale. Studies with scores of less than 6 points were considered low-quality studies, whereas those with scores equal to or greater than 6 points were considered high-quality studies (scores of 6–7 indicate good quality and 8–10 indicate excellent quality).[20]

The GRADEpro GDT online tool was used to evaluate the level of evidence quality of the outcome indicators. The tool is available at its official website http://www.guidelinedevelopment.org/. The GRADEpro GDT online tool for evaluating the quality of outcome indicators includes five degrading factors, namely, risk of bias, inconsistency, indirectness, imprecision and other considerations.[21] The quality of evidence can be divided into four levels, namely, 'high', 'moderate', 'low' and 'very low'.[22]

## Risk-of-bias assessment of individual studies

The quality of the included studies was evaluated and their scores were compared in a consensus meeting between two independent authors (J-LP and LW) to minimise errors and potential biases in the evaluation. However, in the event of any disagreement, a third author (A-LC) was included in the discussion for a final consensus. The Cochrane risk-of-bias 2.0 tool was used to assess the articles' risk of bias.[23] Each article was assessed for selection bias (random sequence generation and allocation concealment), performance bias (blinding of participants and personnel), detection bias (blinding of outcome assessment), attrition bias (incomplete outcome data reporting) and reporting bias (selective outcome reporting). Each domain was rated as high risk of bias, unclear of bias or low risk of bias. The risk map of the biases of the studies' quality was prepared with Review Manager V.5.3.

## Patient and public involvement

No patient participated in writing the system review plan. However, the results were disseminated to patients with SCI.

## Statistical analysis

A meta-analysis was conducted using Review Manager V.5.3. Heterogeneity between studies was evaluated based on the $I^2$ statistic for the quantification of the proportion of the total outcome attributable to variability among studies. The following ranges were defined: $I^2$=0%–30% (no heterogeneity), $I^2$=30%–49% (moderate heterogeneity), $I^2$=50%–74% (substantial heterogeneity) and

$I^2$=75%–100% (considerable heterogeneity).[24] Based on heterogeneity, a random-effects model was used when $I^2$>30%, and a fixed-effects model was used when $I^2$=0%–30%.

For the comparison of data from different scales, pooled statistics were calculated using standardised mean differences (SMDs). Furthermore, means and SDs after intervention and follow-up evaluation for the RAGT and control groups (when relevant) were applied to compute SMDs.

### Addressing missing data

Regarding missing data, the original author was contacted for additional information. In the absence of a reply, the data was calculated based on the availability factor. The potential effect of the missing data on meta-analysis results was tested through sensitivity analysis.

### Subgroup analysis

Analysis results showed a situation wherein heterogeneity was high and subgroup analysis was required. Grouping analysis was conducted based on age (children, adolescents, middle-aged and elderly), SCI level (cervical, thoracic and lumbar), disease course (recovery and sequelae), treatment prescription and treatment duration to address potential heterogeneity and inconsistency. A meta-analysis was also conducted to explore possible sources of heterogeneity.

### Sensitivity analysis

Sensitivity analysis was conducted on the main results to assess the effect of method quality, research quality, sample size, missing data and analysis methods on the results of this review to verify the robustness of the research conclusion.[25]

### Assessment of publication bias

Each included study was evaluated based on the PEDro scale. Funnel charts were used to assess the publication bias of the main results included in the study. However, when the funnel chart was asymmetrical, attempts were made to explain its asymmetry.[26]

### DISCUSSION

RAGT can improve the walking ability of patients with incomplete SCI and can be used by patients with stable vital signs. For patients with complete SCI, RAGT primarily acts to maintain the range of motion of joints. In recent years, there is an increasing number of studies on using RAGT to improve walking ability in SCI, and the new exoskeleton robot for lower limb rehabilitation has shown the advantage of safe transfer. Our current query shows that our work is the first systematic review and meta-analysis on RAGT for patients with SCI. The results of this meta-analysis could help patients and therapists select the appropriate treatment method for SCI and improve new options based on the comparative evidence for effectiveness and safety. Therefore, we hope that the results of this study will provide evidence for guideline recommendations.

### Study limitations

Articles published in both Chinese and English were included. Articles in other languages were not included, and their exclusion may affect our research. When incorporating outcome indicators, all data were sourced from scale evaluation and gait analysis instruments. The lack of research results on neural mechanisms may have had a certain effect on this study.

**Contributors** LW and J-LP, as the first authors, have made equal contributions to this work. Research concept and design: LW and A-LC. Data acquisition: LW and J-LP. Draft: LW and J-LP. Supervised by: A-LC. All the authors approved the publication of this protocol.

**Funding** The authors have not declared a specific grant for this research from any funding agency in the public, commercial or not-for-profit sectors.

**Competing interests** None declared.

**Patient and public involvement** Patients and/or the public were not involved in the design, or conduct, or reporting, or dissemination plans of this research.

**Patient consent for publication** Not applicable.

**Provenance and peer review** Not commissioned; externally peer reviewed.

**ORCID iD**
Lei Wang http://orcid.org/0000-0003-0424-8348

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
