## [Reviewer comments · BMJ Open]

ARTICLE DETAILS

TITLE (PROVISIONAL)	Effect of robotic-assisted gait training on gait and motor function in spinal cord injury: a protocol of a systematic review with meta-analysis
AUTHORS	Wang, Lei; Chen, Lian; Peng, Lin

VERSION 1 – REVIEW

REVIEWER	Wei, Quan Sichuan University West China Hospital
REVIEW RETURNED	27-Jan-2023

GENERAL COMMENTS	1. This manuscript is the first design of a Meta-analysis to assess the efficacy and safety of robot-assisted gait training applied to patients with spinal cord injury. The overall design is reasonable. 2. Search databases and search terms are relatively comprehensive, but the Search databases can also include the Cochrane Central Register of Controlled Trials. 3. In terms of data extraction, the extracted data included information on study type, basic patient information, inclusion exclusion criteria, and intervention such as time, frequency, and intensity etc. For missing information, they took the approach of contacting the author. In the absence of a reply, they calculate available factors, and analyzed the impact of missing data on the Meta-analysis results by sensitivity. Because it was the protocol of Meta-analysis, the specific informations of treatment-related interventions were not mentioned. 4. the statistical software used and some basic statistical descriptors are not described.
--

REVIEWER	Xue, Xiali Chengdu Sport University, School of Sports Medicine and Health
REVIEW RETURNED	04-Mar-2023

GENERAL COMMENTS	1. The references are too few and not new enough. 2. Languages need to be modified by native speakers. 3. What's new about this scheme and previous studies like this one, which have already been published? 4. The Introduction and Discussion are too little and need to be improved. 5. What does TSCI mean, which is only abbreviated. 6. Please upload Preferred Reporting Items for Systematic Reviews and Meta-Analyses Protocols Guideline and Cochrane Collaboration. 7. Please revise and improve the manuscript in detail.
---

VERSION 1 – AUTHOR RESPONSE

Reviewer: 1

According to the suggestion of the reviewer(professor), we have add the Cochrane Central Register of Controlled Trials in Search databases.

Reviewer: 2

1. The references are too few and not new enough.

According to the suggestion of the reviewer(professor), We have added 10 references from the past three years, increasing the number and novelty of references..

2. Languages need to be modified by native speakers.

According to the suggestion of the reviewer(professor), we found professional editors in our research group to make further revisions to the writing of the manuscript, in order to improve the writing quality of the manuscript.

3.What's new about this scheme and previous studies like this one, which have already been published?

Before preparing the writing of the plan, a preliminary search was conducted. The search results revealed several similar studies:

1. The title of the latest article: Effectiveness of robotic assisted gait training on cardiovascular fitness and exercise capacity for incomplete spinal cord injury: A systematic review and meta analysis of random controlled trials, mainly focusing on the study of cardiopulmonary function of patients with incomplete spinal cord injury. The study of patients and observation results are different from our team.

2. Title of the second article: Walking speed is not the best output to evaluate the effect of robotic assisted gain training in people with motor incomplete Spinal Cord Injury: A Systematic Review with meta-analysis.

3. Title of the third article: Is body weight supported treadmill training or robotic assisted gait training superior to surround gait training and other forms of physiotherapy in people with spin cord injury? A systematic review。 This study was published in 2017, which was earlier and included less literature. In the end, a systematic review was conducted without meta-analysis, and intervention factors included robots and weight loss treadmills.

In summary, our research mainly focuses on exploring the effects of robots on gait and motor function in patients with spinal cord injury, which is not covered by other studies and is also an innovative point of our research.

4.TheIntroduction and Discussion are too little and need to be improved.

According to the suggestion of the reviewer(professor), Some modifications have been made in the introduction section, as the article type is a protocol, which is limited in the introduction and discussion section, making it impossible to conduct in-depth discussions, resulting in a lack of all content.

5.What does TSCI mean, which is only abbreviated.

Thank you very much for the questions raised by the reviewers. After careful inspection by the team, it has been confirmed that the letter T in the abbreviation (TSCI) was caused by a clerical error, and the correct one is SCI. Thank you again. The team members have carefully reviewed the article and should prevent such issues from occurring.

6.Please upload Preferred Reporting Items for Systematic Reviews and Meta-Analyses Protocols Guideline and Cochrane Collaboration.

According to the suggestion of the reviewer(professor), We have supplemented the PRISMA-P checklist and indicated the line number of your manuscript.

7.Please revise and improve the manuscript in detail.

According to the suggestion of the reviewer(professor), our team has further improved and revised the article, including multiple parts such as the introduction and methods.

Thanks again for the comments of experts on the manuscript!

VERSION 2 – REVIEW

REVIEWER	Xue, Xiali Chengdu Sport University, School of Sports Medicine and Health
REVIEW RETURNED	19-Apr-2023

GENERAL COMMENTS	1. It should clear the methods and higherchy of the evidence and the level also.2 Were retrospective trials included? If not, make it clear in the abstract.3. Applied keywords used in the searching process have to be clearly described in the abstract.4. Regarding the statistical analysis section, subgroup analysis should be described in much more detail.5. "Line 152-153. The risk map of the biases of the studies' quality was prepared with RevMan 5.2 software. Line 158. A meta-analysis will be conducted by using Review Manager 5.3." Please explain why Revman is not the same version6. The discussion is too short and not deep enough. Please focus on the research for further detailed elaboration.7. What are the limitations of the study? Please add.
--

VERSION 2 – AUTHOR RESPONSE

Reviewer: 1

1. It should clear the methods and higherchy of the evidence and the level also.

Reply: In response to the reviewer's first comment, our research team is not very clear about which part of the content it refers to. If it refers to the suggestions for the method section, our research team has made modifications based on the reviewer's first comment.

2 Were retrospective trials included? If not, make it clear in the abstract.

Reply: Excluding retrospective trials. According to the suggestion of the reviewer, additional explanations have been provided in the abstract section of the manuscript

3Applied keywords used in the searching process have to be clearly described in the abstract.

Reply: According to the suggestion of the reviewer. The keywords used in the search process have been supplemented in the abstract section of the manuscript.

4.Regarding the statistical analysis section, subgroup analysis should be described in much more detail.

Reply: According to the suggestion of the reviewer. subgroup analysis had be described in much more detail in the subgroup analysis section of the manuscript.

5.“Line 152-153. The risk map of the biases of the studies’ quality was prepared with RevMan 5.2 software. Line 158. A meta-analysis will be conducted by using Review Manager 5.3.” Please explain why Revman is not the same version

Reply: Firstly, we would like to thank the reviewer’s comment, which was a mistake made by our research group in writing. In the early days, our software came with Review Manager 5.2 , but later we upgraded to Review Manager 5.3 and forgot to make any changes. We will ultimately use Review Manager 5.3 in the article and have made modifications in the manuscript. Thank you again.

6.The discussion is too short and not deep enough. Please focus on the research for further detailed elaboration.

Reply: Dear reviewer, there have been no significant adjustments made by the research team regarding this comment. Due to the research is a protocol and the lack of research data and results, it is not possible to discuss and elaborate on the article in the discussion section. Thank you for the reviewer's suggestions.

7. What are the limitations of the study? Please add.

Reply: According to the suggestion of the reviewer. The limitations of the study in the discussion section of the manuscript.

Thank you again for the comments provided by the reviewers!